# Methods, Thermodynamic Applications, and Habitat Implications of Physical and Spectral Properties of Hair and Haircoats in Cattle

**DOI:** 10.3390/ani13193087

**Published:** 2023-10-03

**Authors:** Kifle G. Gebremedhin, Vinicius D. F. C. Fonseca, Alex S. C. Maia

**Affiliations:** 1Department of Biological and Environmental Engineering, Cornell University, Ithaca, NY 2503, USA; 2Animal Biometeorology Laboratory, São Paulo State University, Jaboticabal 14884-900, SP, Brazil; carvalho.fonseca@unesp.br (V.D.F.C.F.); alex.maia@unesp.br (A.S.C.M.)

**Keywords:** cattle haircoat, physical and spectral properties, methods of measurement, thermodynamic models, heat exchange, habitat implications of haircoat color

## Abstract

**Simple Summary:**

The physical properties (hair diameter, hair length, haircoat depth, and haircoat density) and spectral properties (solar absorptivity, solar reflectivity, and solar transmissivity) of hair and haircoats play critical roles in heat and moisture exchange between an animal and its surrounding environment. These properties also play an important role in protecting the skin against penetration of ultraviolet radiation. Black haircoats absorb solar radiation, but the absorption occurs at the haircoat–air interface (away from the skin surface) where convective heat loss is high. Holstein cows with dominant black color haircoats are more suitable in latitudes with high solar input than those with a white color haircoat. A white haircoat is more transparent and allows solar energy to penetrate deeply into the haircoat, and thus, heat flows toward the skin surface (heat gain). The physical properties of hair and haircoats are not numerically the same at different locations (dorsal, ventral, lateral, neck, head, etc.) of the body of a cow. The density (no. of hairs/cm^2^) of a haircoat is constant to a certain depth from the skin surface, and then decreases exponentially toward the haircoat–air interface. Cattle with a dominant black haircoat spend more time using shade than those with a white or red haircoat.

**Abstract:**

The physical properties (hair diameter, hair length, haircoat depth and haircoat density) and spectral properties (absorptivity, reflectivity, transmissivity) of the hair and haircoat of cattle are inputs to heat and moisture exchange between the skin surface and the surrounding environment, and thus play a critical role in body temperature regulation. Physical and spectral properties of haircoats also play an important role in protecting the skin against penetration of ultraviolet radiation. The focus of this review is to identify accurate and consistent measurement procedures of these properties. Additionally, the paper shows the utilization of the properties on heat exchange models and their implications on voluntary thermoregulation of cattle. To highlight the effects and benefits of haircoat color vis-à-vis solar radiation and its implication on ecological habitation, a brief explanation is provided using polar bears (white haircoat in a cold environment) and black goats in a hot desert environment.

## 1. Introduction

The haircoat of animals plays a critical role in heat and moisture transfer from the skin surface to the surrounding environment, and in body temperature control. Haircoats also protect the skin against penetration of ultraviolet radiation [1]. Endotherms interact with the thermal environment and affect their growth, production, and reproduction potential at the skin–haircoat interface [2]. Animal haircoats trap air to provide insulation and thus conserve body heat (essential during cold weather) but become an obstruction for evaporative cooling from the skin surface by reducing the velocity and moisture gradients through the fur layer in hot and humid conditions [2]. When an animal is exposed to direct sunlight, a temperature gradient is formed between the haircoat surface and the skin [3], which, in turn, depends on haircoat depth. Cattle adapted to tropical regions have low haircoat thickness (~3 mm), which results in a small thermal gradient between the haircoat and skin surface [4].

A haircoat is one of the most significant factors that affects the heat dispersion rate from the body to the environment [5]. Heat flow occurs through processes dependent on the conditions of the surrounding environment, such as humidity and temperature. In hot and humid areas, animals must be able to dissipate heat through the skin and from the respiratory tract and lose thermal energy coming from the environment at the same time. The protective properties largely depend on the physical characteristics of the haircoat and skin, which allow for the exchange of heat with the environment [6]. Various haircoat characteristics are greatly involved in heat transfer between the skin surface and the surrounding environment. For example, a light-colored and shiny haircoat reflects a greater proportion of solar radiation than a dark-colored and wooly haircoat [6]. The rate of radiant heat exchange between the skin surface and the surrounding environment largely depends on the physical characteristics of the haircoat (hair length, hair diameter, color, haircoat depth, and haircoat density) and skin color [6]. The same properties also influence behavioral thermoregulation, panting, and sweating [7]. For example, a light-colored and shiny haircoat reflects a greater proportion of incident solar radiation than a dark-colored and wooly haircoat [6]. The *Bos indicus* cattle (e.g., Nellore, Guzerat) with a light-colored haircoat over a well-pigmented skin are well adapted to tropical regions, where solar irradiance is high and almost constant throughout the year [6]. Conversely, the dominant black Holstein cattle are better adapted to survive and to produce in these regions than the white-haired ones, as they possess a skin with greater levels of pigmentation [6]. Cattle acclimate to their environment, but at a cost. Collier and Gebremedhin [8] reported that the process of acclimation results in lower outputs of domestic animals, and this loss in production is more pronounced in higher-producing cows.

Haircoat properties are inputs to heat and mass transfer models of animals. It is, therefore, imperative that physical and spectral properties of haircoats are accurately characterized for modeling heat and mass transfer through the haircoat. Moreover, haircoat properties can provide information about the breed of animals that are more likely to adapt to a particular environment. The availability of physical and spectral data in the literature is sparse at best; what is available is mostly from studies done in the early 1950s to 1970s, which do not represent present-day high-milk-volume-producing cows and changes in nutrition of animal feed. The objectives of this review are:(1)To compile physical and spectral properties’ data about hair and haircoats that represent present-day cattle that are available in the literature;(2)To identify methods and procedures for measuring hair and haircoat properties so that a common approach or standard could be followed for future research;(3)To show how the physical and spectral properties are applied in models of heat exchange between an animal and its surrounding environment, and their implications in behavioral thermoregulation of cattle.

## 2. Literature Review

### 2.1. Physical Properties of Haircoats

The availability of physical and optical data in the literature is sparse at best; what is available is mostly derived from studies conducted in the early 1950s to 1970s. One of the earlier studies was by Hutchinson et al. [9], who measured physical properties of Friesian cows of different colors and coat characteristics. They divided cow hair into three different colors, such as white, brown, and black. More deeply, they subdivided hair colors into several groups by how dense hairs are aligned on the skin. They reported that haircoats of brown color had longer haircoat depth and a heavier weight of haircoat per square centimeter than white and black haircoats. Brown haircoats were two to three times thicker than white haircoats and about seven times thicker than black haircoats. For weight of haircoat per square centimeter, white haircoats and black haircoats showed similar weights, but brown haircoats were twice as heavy per square area than white and black haircoats. White haircoats and black haircoats had greater haircoat density than brown haircoats.

Peters et al. [10] measured physical properties of haircoats of different breeds of cows. They studied raw and clean hair weights per square centimeter of different breeds and crossbreeds of cows. The Cattalos were predominantly one fourth Bison, with an average of 23% of Bison parentage. Hair samples were obtained from the left mid-rib section of 150 cm^2^ of each cow. They reported data of raw weight and clean weight of haircoats for different breeds of cows. Pure breed cows such as Hereford, Angus, and Shorthorn have less raw and clean hair weight than those of crossbred cows. The hair weight of Hereford x Bison showed the greatest raw and clean hair weight among all other breeds, two and three times heavier than others. Moreover, they compared the raw and clean hair weight at different body locations and various breeds. The data show that the raw and clean hair weight of Bison were about three times heavier than those of Hereford and Cattalo. Both bulls and heifers had relatively similar hair length and hair diameter, yet heifers had a greater haircoat density than bulls. There was no significant difference observed in hair length and diameter for different breeds of cows. Clean weight and haircoat density of Domestic X Bison cows are much greater than those of other breeds of cows.

More recently, Gebremedhin et al. [11] measured the physical properties of hair and haircoats from four breeds of heifers with different haircoat colors. The breeds were Angus (black), MARC III (dark red), MARC I (tan), Charolaise (white). MARC III is crossbreed of ¼ Pinzgauer, ¼ d Poll, ¼ Hereford, and ¼ Angus, and MARC I is crossbreed of ¼ Charolaise, ¼ Braunvieh, ¼ Limousin, 1/8 Angus, and 1/8 Hereford. The results of the study are given in Figure 1a–d. Gebremedhin et al. [12] also measured the physical properties of hair and haircoats of Holstein calves. The haircoat samples were collected from the rump, lumbar, belly, thoracic, and head areas of three Holstein calves. The density of a haircoat is constant up to a certain haircoat depth from the skin surface and then decreases exponentially. For example, the density at the skin level of the lumbar area was about 7200 hairs/cm^2^, but beyond 1.3 cm away from the skin surface, at the haircoat–air interface, the density decreased exponentially, as shown in Figure 2a. Similarly, the density at the ventral (belly and lateral) was about 6200 hairs/cm^2^ up to 1.3 cm away from the skin surface and then decreased significantly, as shown in Figure 2b,c. Haircoat density is, therefore, constant until the length of the shortest hair. The mean and standard deviation of the measured properties are given in Table 1. Haircoat density is expressed by a hyperbolic tangent function as
(1)ρ(z)=4531∗(1−tanh(z−2.0)−3499)

where *z* is the depth (mm) of the haircoat originating from the haircoat–air interface, and *ρ* is density of the haircoat (number of hairs, mm^−2^).

Bertipaglia et al. [3] measured physical properties of hair and haircoats of Holstein cows. The cows were from a commercial dairy herd in Descalvado, State of Sao Paulo, Brazil. They used 939 Holstein cows in the study, and sampling was carried out from November 2000 to April 2001. Hair length was measured from the 10 longest hairs from the sample, and haircoat density was measured by visual direct counting. These data are also given in Table 1. Bertipaglia et al. [6] also measured haircoat depth, density of haircoat, hair length, and hair diameter of Braford cows from 1607 Nelore-Hereford crossbred females born between 1998 and 2002 from a commercial Bradford herd in Navirai, State of Mato Grosso do Sul, Brazil. This region has a tropical humid weather, with a hot and wet summer and dry winter. Hair samples were obtained from late October to December, 2003. The depth of the haircoat was measured at the center of the thorax, about 20 cm below the dorsal line, and hair samples were collected from the same location. The density of the haircoat was measured by direct counting of hairs in the sample. The data are also given in Table 1.

Da Silva et al. [14] measured haircoat properties of five breeds of cows: Holstein, Nelore, Canchim, Simmental, and Brangus. Samples were taken from the central area of the trunk, about 20 cm below the spinal column of adult cattle. Haircoat depth of the Nelore and red Holstein was two times thicker than that of the other breeds. Brangus, white Simmental, and Cranchim had longer hair length (17 to 20 mm) than the other breeds. The Nellore cows had the highest haircoat density (2083 hairs/cm^2^) and thickest hair diameter (54.0 µm). The data of this study are given in Table 1. Similarly, Verissimo et al. [13] measured physical properties of haircoats of three genetic groups of cows, namely, Gyr, crossbred Gyr x Holstein with Holstein fraction between 50 and 75%, and crossbred Gyr x Holstein with Holstein fraction above 75%. The measurements were conducted at Ribeirao Preto, State of Sao Paulo, Brazil, where the climate was classified as a tropical savanna, with a rainy summer and a dry winter. Hair samples were collected from the scapular region of each cow during the summer of 1994. The data are summarized in Table 1. In summary, the physical properties of hair and haircoats vary with breed, location in the body, and climate. Cows in tropical climates (Brazil) have shorter hair length and lower haircoat density than similar cows in winter climates in the U.S.A. According to Maia et al. [15], there seems to be no significant variation in haircoat properties of Holstein cows with age, and hair diameter was slightly thicker in December. Gilbert et al. [16] used least-square means and standard errors to analyze haircoat characteristics. They reported that no significant difference in the physical properties of haircoats was observed in different types of breeds, age, and sex. However, the undercoat had a greater weight and density and shorter hair length and diameter than the guard hair.

The physical properties of haircoats also affect cattle’s resistance to other environmental stressors such as tick infestation and tick-induced diseases [17,18,19]. These studies reported that cattle breeds with thicker coats and long hairs are more susceptible to tick infestation than those with shorter hairs. This haircoat characteristic is more likely to be expressed in *Bos taurus* cattle breeds. In hot–humid tropical conditions, Verissimo et al. [13] observed that crossbred-between-Gyr-and-Holstein cattle were more likely to be infested by ticks than purebred *Bos indicus* Gyr cows. This higher susceptibility was associated with crossbred animals with thicker haircoats. Ibelli et al. [17] compared resistance to tick infestation and its association with haircoat characteristics between crosses of Senepol x Nellore, Angus x Nellore, and purebred Nellore cattle. They found that the Nellore cattle and the crossbred Senepol x Nellore had lower haircoat thickness and were less likely to be infested by ticks than the crossbred Angus x Nellore. Similarly, in a semi-arid rangeland of South Africa, Murufu et al. [18] observed that a Nguni short-haired cattle, a hybrid *Bos indicus* x *Bos taurus* naturally adapted to the semi-arid rangelands of South Africa, were less susceptible to tick infestation than an exotic thicker-haired *Bos taurus* Bonsmara cattle. The mechanisms underlying this association are still poorly understood. A thicker haircoat can possibly provide a better protection to ticks against predators (e.g., birds) and from the animal’s self-grooming. However, the genetic resistance of cattle to tick infestation and tick-induced diseases seems to be more dependent on skin immune responses and abundance of tick-resistant genes [19]. In a recent study, Feltes et al. [20] reported a very low genetic correlation between tick count and haircoat characteristics in a population of crossbred Angus x Nelore, which suggests that these traits are regulated by different genes.

### 2.2. Spectral Properties of Haircoat

Transmissivity is the fraction of irradiation transmitted through the haircoat toward the skin surface, reflectivity is the fraction of incident energy reflected back from the haircoat and skin surface together, and absorptivity is the fraction of irradiation absorbed by the haircoat. Many investigators [7,11,14] have measured spectral properties of haircoats. The properties of interest are transmissivity, reflectivity, and absorptivity. Gebremedhin et al. [11] measured reflectivity and transmissivity, and using Equation (2), calculated the absorptivity of the haircoat of four breeds of heifers. The results are shown in Figure 1 and Figure 3. The average absorptivity values are given in Table 2, and the percent reflectivity expressed as a function of wavelength is shown in Figure 4. Hillman et al. [21] measured the absorptivity of black and white hair of lactating Holstein cows. The data are given in Table 2. They used a pyrometer to measure the incident and reflected short-wave radiation.

Maia et al. [22] measured optical properties of Holstein cows. A total of 973 female Holstein cows were used in the study. It was conducted in a commercial herd in Descalvado, State of Sao Paulo, Brazil. An LI-1800 spectro-radiometer was used to measure the reflectivity and transmissivity, with wavelengths ranging from 300 to 850 nm. Absorptivity was calculated by subtracting reflectivity and transmissivity from 1.0 (Equation (2)). The data are given in Table 2, and the relationship between absorptivity and wavelength is shown in Figure 5a,b [14]. Da Silva et al. [14] measured the reflectivity and absorptivity of haircoats for different breeds of cows and haircoat colors. They used an LI-1800 spectro-radiometer at wavelengths ranging from 300 to 850 nm. The data are given in Table 2. In summary, they found that darker haircoats have lower reflectivity than brighter haircoats. Maia et al. [22] also measured the transmissivity and absorptivity of Holstein cows in different months of sampling for different cow age and different coat color. The data are given in Table 3. There was no difference in effective absorptivity due to month of sampling. The range was between 0.84 and 0.85. A large difference was observed in transmissivity due to month of sampling. The transmissivity was larger from November to February than that from March to April. The transmissivity in cold weather is about two to three times larger than that in hot weather. Effective absorptivity was not affected by age. The transmissivity was high for cows below 2 years of age and between 6.5 and 8 years of age. Transmissivity was lowest for cows between 2 and 3.5 years of age.

Transmissivity of UV radiation was lower in mixed-colored haircoats (white and grey hairs) compared to black, red, and white haircoats Figure 6 [1,15]. A mixed-colored haircoat and black skin provided the best protection against UV penetration. This type of haircoat is found in *Bos indicus* Nellore and Guzerat cattle. A white or red haircoat and rosy skin, which is typically found in white spots of Holstein cattle, had the highest transmissivity of UV radiation [1,15]. This fact explains the skin damages and neoplasia (Figure 7 [23]) in the predominantly white and red Holstein cows in the tropical regions. In the equatorial tropical areas, where dairy cattle are mostly in pasture, the haircoat color of the Holstein cattle is more predominantly black than white. The radiative characteristics of haircoats are influenced not only by the spectral properties, but also by the physical properties. Cena and Monteith [24] proposed the concept of effective reflectivity (α*), effective absorptivity (*ρ**), and effective transmissivity (τ*), which can be calculated as follow:(2a)$$\rho *=\frac{\left[\rho_{c}\left(1-\rho_{s}\right)-\alpha_{c}\;\rho_{s}\right]sinhx+k\rho_{s}coshhx\;}{\left[\rho_{c}\left(1-\rho_{s}\right)+\alpha_{c}\;\right]sinhx+kcoshhx}$$
(2b)$$\tau *=\frac{k\;}{\left[\rho_{c}\left(1-\rho_{s}\right)+\alpha_{c}\;\right]sinhx+kcoshhx}$$
(2c)$$\alpha *=1-\rho *-\tau *\left(1-\rho_{s}\right)\;$$
(2d)$$k={\left[\alpha_{c}^{2}+2\rho_{c}\alpha_{c}\right]}^{0.5}\;$$
(2e)x=kTP 
(2f)$$P=ND\;\left\{tan\;tan\;\left[arccos\;arccos\;\left(\frac{T}{L}\right)\;\right]\;\right\}$$
where sinh *x* and cosh *x* are the hyperbolic sine and cosine of *x,* respectively; *P* is the fraction of radiant energy intercepted by a unit depth of haircoat; *N* is the density of the haircoat (hairs/m^2^); D is the average hair diameter (m); T is the coat depth (m); and L is the average hair length (m). The *ρ*_c_ and *α*_c_ are the respective values for reflectivity and absorptivity of the haircoat, and *ρ*_s_ and *α*_s_ are reflectivity and absorptivity of the skin. According to the results published by Maia et al. [22], the effective transmissivity was greater in less dense coats with short and thin hairs. The effective reflectivity was influenced more by the variation in the spectral properties of the haircoat and skin rather than the physical properties of the hair. The effective absorptivity is likely to be greater in black and dense coats with long and thick hairs than in white and less dense coats with short and thin hairs. In terms of seasonal changes, the effective transmissivity was larger from November to February than that from March to April.

## 3. Measurement Procedures

### 3.1. Physical Properties

The physical properties are haircoat density, hair diameter, haircoat depth, and hair length.

#### 3.1.1. Haircoat Density

The following procedures were used to count haircoat density from animal pelts:Prepare samples from several locations (such as rump, lumbar, belly (ventral), thoracic (lateral), and head) of the animal;Cut hairs at ≤1 mm from the skin layer. Blow the shaved part with a stream of air jet to remove stray pieces of hairs. Clip all hairs from an area large enough to permit total counts of more than 20 squares on an ocular grid (e.g., 100 hairs/cm^2^);Adjust lighting to permit counting of individual hairs in stubble. Very intense light at nearly parallel to skin helps if skin and hairs are both black;Count hairs on a random number chart built into the ocular grid;Calibrate the ocular grid using a stage micrometer or using a good ruler.

#### 3.1.2. Hair Diameter

There are two ways to measure hair diameter. One method is to use a digital Vernier Caliper and the other method is to use a calibrated ocular micrometer. The procedure for the latter is as follows:Clip off hairs at ≤1 mm from the skin layers with aris scissors and put the hairs in a microscope slide with dissecting forceps. Apply a drop of water on the slide to pressure the hairs;Separate individual hairs in rows;Measure diameters under the calibrated ocular micrometer.

#### 3.1.3. Haircoat Depth

Haircoat depth can be measured with a digital Vernier Caliper either from live animals at different locations of the animal body or from frozen pelt samples. It is necessary to take repeated measurements to obtain good statistics. It is important that the caliper end does not depress the skin. That is why it would be necessary to freeze the pelt.

#### 3.1.4. Hair Length

A rough hair length can be measured with a ruler, but if an accurate measurement is desired, it should be done using a dissecting microscope. The procedures for the latter method are as follows:Under a dissecting microscope, clip hairs at less than 1 mm from the skin surface with an iris scissor or equivalent;The hair will stick to the scissor. Put the scissor tip with hairs on it in a drop of soap and water or “Tame” cream rinse, which is on a microscope slide. The soap may be any liquid type that will act as a wetting agent. Do not put more than 20 hairs on a slide. Separate the hairs by spreading them around and put them in rows so that they will be traceable individually;Cover the hairs with cover slip and seal layers with Permount (mounting medium) to keep the water from evaporating (water is to preserve the hairs);Measure the length of the hair with a camera lucida and a map-measuring device. The map-measuring device should be calibrated under the camera lucida using a ruler. Hair length is measured by tracing the image with the map-measuring device. A total sample size of about 200 hairs may be needed for a given body location.

Use the map measurer and camera lucida setup to measure the image of the stage micrometer.

### 3.2. Spectral Properties of Haircoat

The spectral properties of interest are transmissivity, denoted by τ, reflectivity, denoted by ρ, and absorptivity, denoted by α.

#### 3.2.1. Transmissivity and Reflectivity

Transmissivity and reflectivity can be measured using a Beckman DK-2A ratio recording spectrophotometer (Figure 8 [12]), an LI-1800 spectro-radiometer [14,15,22], or using solar simulator [1]. The device simulates solar radiation at wavelengths between 250 and 360 nm.

#### 3.2.2. Absorptivity

The absorptivity coefficient can be calculated from the relationship:(3)α=1.0−(ρ+τ)

## 4. Applications of Hair and Haircoat Properties in Thermodynamic Models

As mentioned previously, hair and haircoat properties are inputs to models of heat flow through the haircoat. This section will illustrate how and where the properties are applied. The focus is not on the solution of the energy equations but rather on how the properties are applied in the equations. The governing equation of heat flow through the haircoat for a steady-state heat conduction with a heat-generation term can be expressed as
(4)$$k_{eff}\frac{d^{2}T}{dx^{2}}+q_{sol}^{\prime \prime }=0$$
where k_eff_ is the effective thermal conductivity of the haircoat, and $q_{sol}^{\prime \prime }$is the volumetric heat-generation term within the haircoat due to absorption of solar radiation. The boundary conditions needed to solve the equation are (1) at the haircoat–-air interface, the conduction term is equal to the convective term, and (2) at the skin surface, the temperature is equal to a specified skin temperature. The two boundary conditions can be mathematically expressed as follows:

at the haircoat–air interface (x = 0):(5a)$$k_{eff}\frac{dT}{dx}=h_{c}(T-T_{air})$$
at the skin surface (x = L):T = T_skin_(5b)
where h_c_ is the convective heat transfer coefficient at the haircoat–air interface, and L is depth or thickness of the haircoat. The effective conductivity, *k_eff_*, in the conduction term, which accounts for the physical properties of the haircoat, is a porous medium and is calculated as
(6)$$k_{eff}=\varepsilon k_{air}+(1-\varepsilon )k_{hair}$$
where ϵ is the void fraction and k is the thermal conductivity of air (*k_air_)* and hair *(k_hair_*). The void fraction of the haircoat can be calculated using an equation as [25]
*∈* = (1−*A_h_*/*A_t_*)(7)
where A_h_ is area of the haircoat, and A_T_ is the total area of the haircoat and air.

To simulate the effects of radiation within the haircoat, the equation developed by Cena and Monteith [24] determines the average penetration per unit depth of radiation in the haircoat and is expressed as
(8)$$z_{avg}=2\times \rho_{h}\times D\times tan\;\left[a\;cos\;\left(\frac{l}{L}\right)\right]$$
where *z_ave_* is the average depth to which the radiation penetrates the coat (mm^−1^), ρ_h_ is haircoat density (number of hairs, mm^−2^), D is hair diameter (mm), l is depth of the haircoat (mm) and *L* is the hair length.

A more elaborate model of sensible heat loss (q^″^_sen_) through the haircoat can also be determined from an energy balance at the skin surface (z = 0) [26,27,28]:(9)$$q_{sen}^{"}={\left.k_{eff}\frac{\partial T}{\partial z}\right|}_{z=0}-\sigma T_{skin}^{4}-2\sigma T_{sky}^{4}E_{3}(\beta z_{L})-2\sigma \int_{0}^{z_{L}}\beta TE_{2}(\beta z^{*})dz^{*}-q_{sol}^{"}$$
where the terms to the right of the equal sign are, respectively, the conduction component, thermal radiation from the skin surface, radiation due to cold sky, thermal radiation across the depth of the haircoat, and solar absorption within the haircoat. The convection component is included indirectly through modifying the temperature profile across the haircoat. E_2_ and E_3_ are exponential integral functions and are defined by Siegel and Howell [26], z^*^ is a dummy variable of integration, and β (1/m) is an absorption coefficient for thermal radiation and is dependent on hair diameter (D) and haircoat density (*ρ*) and is computed by [27,28]:(10)$$\beta =\frac{2}{3\pi }\rho \times D$$

Haircoat physical properties also affect the heat of evaporation from the skin surface. Fick’s Law [29,30] uses the difference in concentration of water on the skin surface and that at the haircoat–air interface.

## 5. Habitat Implicatons of Hair and Haircoat Physical and Spectral Properties

As illustrated in the above applications, the physical and spectral properties of hair and haircoats are inputs to thermodynamic models that predict energy exchange between the animal and the environment [31,32,33,34]. For example, haircoat density acts as a resistance to wind penetration into the haircoat, and thus affects the boundary condition. Similarly, the higher the density of the haircoat, the more difficult it is for the animal to lose heat by evaporation from the skin surface [35,36]. Thus, haircoat serves as an insulation by trapping air during winter but is an obstruction to evaporative cooling from the skin surface during hot summers.

The spectral data show that hair color determines the level of absorptivity and reflectivity to solar radiation. We did not conduct any study about polar bears at the poles nor about black goats in the desert, but we want to highlight the implications of color of hair and haircoats on animal habitation. Polar bears have a white haircoat color, and white is a good reflector of heat; thus, it seems the polar bears would not be able to take advantage of the sun, when it is out, to get warm. In the desert environment, you find black goats, and black is a good absorber of heat, which would mean that the desert goats would be “cooked” (thermally heat stressed) in the hot desert climate. It seems that Mother Nature has made a mistake by having the polar bears at the poles and the black goats at the desert. Should the polar bears and goats swap locations, or should the polar bears have black haircoats and the goats white? From a camouflaging standpoint, the polar bears should have a white haircoat to disguise themselves from predators, but from a thermal standpoint, a white haircoat is more transparent and allows solar energy to penetrate deeply into the haircoat. As a result, the peak of the temperature profile through the haircoat is deep down into the haircoat, and thus, heat flows both toward the skin and the haircoat–air interface, where it is lost by convection. This way, the polar bears have an advantage when the sun is out. The skin of the polar bear is black, which also enhances the absorption of heat from thermal radiation, as well as provides good protection against the penetration of energy photons from the ultraviolet band. In a black haircoat, the peak temperature is close to the air interface (Figure 9), where the heat loss is high due to convection by wind. Two classic studies conducted by Finch et al. [37] and Shkolnik et al. [38] confirmed this hypothesis. They found that, although the black Bedoiun goats absorb two to three times as much heat by thermal radiation from the sun as the white ones, the black goats had higher rates of convection in the air beneath the coat, which, in turn, transferred the heat to the environment before reaching the skin surface.

In thermally stressful environments, animals seek shade, if available, and those with black haircoats seek shade and spend more time in shade than cows with white-, red-, or tan-colored haircoats (Figure 10). Finch et al. [39] studied three black-haired and three white-haired cattle breeds and provided the first robust evidence on the influence of physical and spectral haircoat traits on the grazing and shade-seeking behavior of cattle. They found that light-haired steers spent more time in the sun and grazing than dark ones. Moreover, cattle breeds with deep and wooly coats also spent less time grazing and sought shade earlier than those with less thick coats. Similarly, in a study conducted in a hot-humid region, dark-haired cows spent more time using shade than those with predominantly white coats [40]. The thermoregulatory costs for a cow to graze during times of high radiant heat load, especially in terms of water use, is higher if she has a black haircoat rather than a white one [41,42,43]. For instance, on a hot, sunny day in a tropical environment, black-haired dairy cows can absorb as much as 600 W m^−2^ of short-wave and long-wave solar radiation, an amount of heat load that would require up to 300 g h m^−2^ of sweat to be evaporated in order to maintain their thermal equilibrium; while a typical, dominant white-haired cattle have the potential to absorb much less heat by thermal radiation, which would result in less requirement for cutaneous evaporative water loss. Although well-hydrated cows can sustain high rates of sweating [44], if shade is available, black-haired cows are likely to seek shade at lower levels of solar irradiance than those with a white haircoat. In terms of habitat implications, when compared with white-haired cattle breeds, even though a black haircoat would represent a disadvantage in terms of radiant heat gain for animals living in tropical hot-sunny areas, animals with a black haircoat have skin that is better protected against the penetration of ultraviolet radiation, which, in turn, makes them less susceptible to developing skin burns and cancer [45]. A recent long-term analysis conducted in tropical regions suggested that Holstein cattle with a 90% black haircoat have greater longevity than Holstein cattle with a 90% white haircoat [46,47]

## 6. Concluding Thoughts

The following key points can be drawn from this review:Data and procedures for measuring and recording physical (hair length, diameter, haircoat thickness and density) and spectral (reflectivity, absorptivity and transmissivity) properties of hair and haircoats of cattle are identified. These properties are inputs to models of thermoregulation of cattle;The physical properties of hair and haircoats are not numerically the same at different locations (dorsal, ventral, lateral, neck, head, etc.) of the body of a cow. The density (no. of hairs/cm^2^) of a haircoat is not constant throughout the depth of the haircoat. It is expressed by a hyperbolic tangent function. The average solar absorptivity of the haircoat of four breeds of cows were black = 98%, dark red = 92%, Tan = 75%, and white = 37%;The effective thermal conductivity of a haircoat increases with hair density and hair diameter. A less dense haircoat presents lower effective transmissivity regardless of the color of the haircoat;The effective absorptivity is greater in cattle with black and dense coats with long and thick hairs than that in white and less dense coats with short and thin hairs. Despite having a higher absorptivity of solar radiation, Holstein cows with predominantly black haircoats are more suitable in latitudes of high solar radiation than those with white haircoats;In hot climates, a black haircoat acts as a solar shield because the peak temperature in the haircoat is close to the haircoat–air interface (away from the skin) where convection heat loss is high because of the wind effect, whereas a white haircoat is more transparent and thus allows solar energy to penetrate deep into the haircoat, thus causing heat to flow toward the skin surface (heat gain);Animal haircoats trap air to provide insulation and thus conserve body heat (essential during cold weather) but become an obstruction for evaporative cooling from the skin surface by reducing the velocity and moisture gradients through the fur layer in hot and humid conditions;The physical and spectral properties of hair and haircoats lead to important thermoregulatory behavior, such as seeking and staying in shade when thermally stressed. Cattle with black haircoats spend more time using shade than those with white and red haircoats.

## Figures and Tables

**Figure 1 animals-13-03087-f001:**
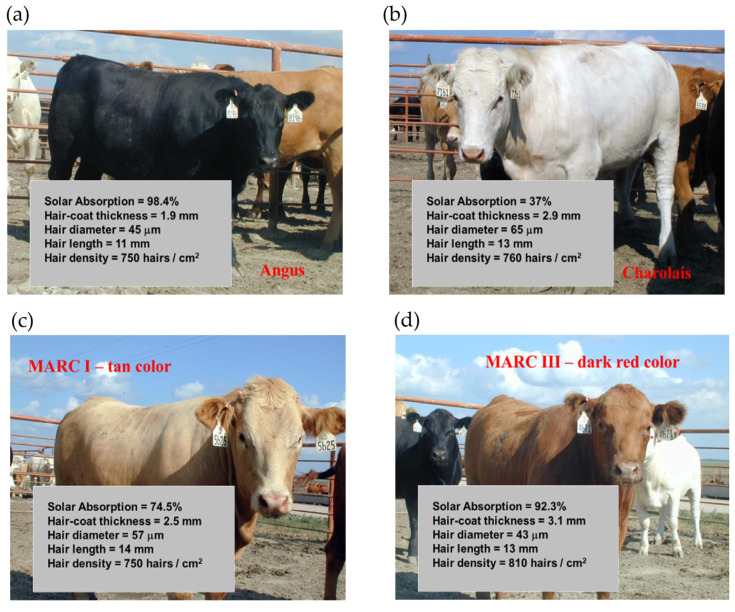
Measured physical properties of hair and haircoat, (**a**) Angus—black color, (**b**) Charolaise—white color, (**c**) MARK I—tan color, and (**d**) MARC III—dark red color [11].

**Figure 2 animals-13-03087-f002:**
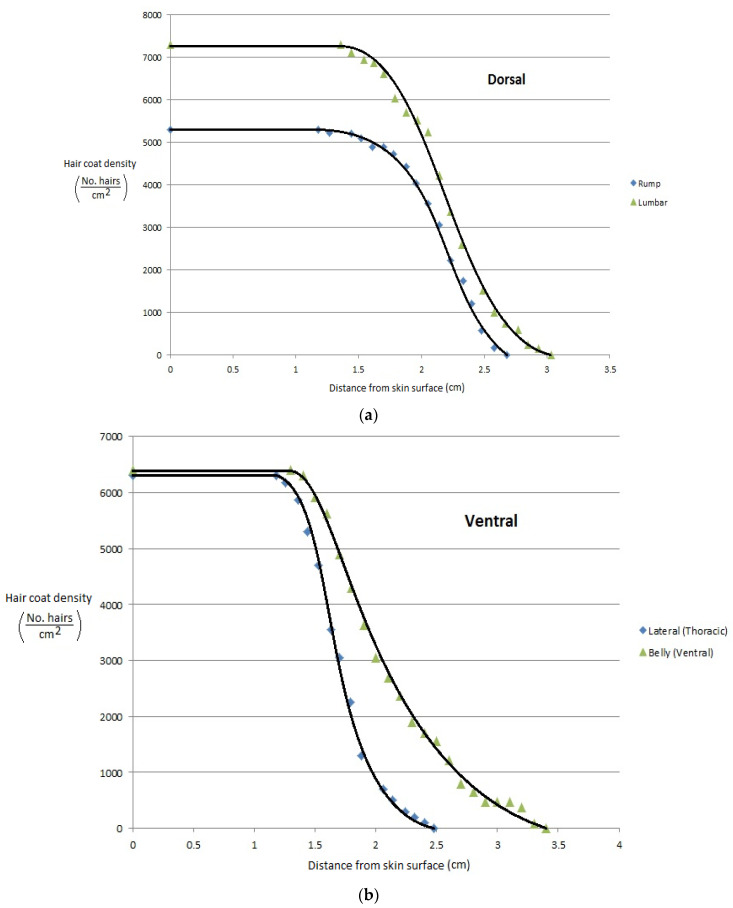
(**a**) Haircoat density profile from the skin surface through the thickness of the haircoat at the rump and lumbar areas [12]. (**b**) Haircoat density profile from the skin surface through the thickness of the haircoat taken at the lateral and belly areas [12]. (**c**) Haircoat density profile from the skin surface through the thickness of the haircoat taken at the neck area [12].

**Figure 3 animals-13-03087-f003:**
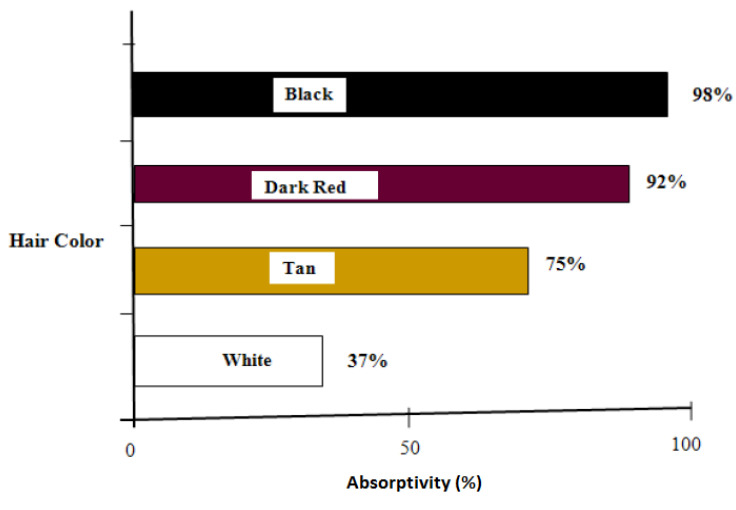
Average absorptivity of the haircoat of four breeds of cows [11].

**Figure 4 animals-13-03087-f004:**
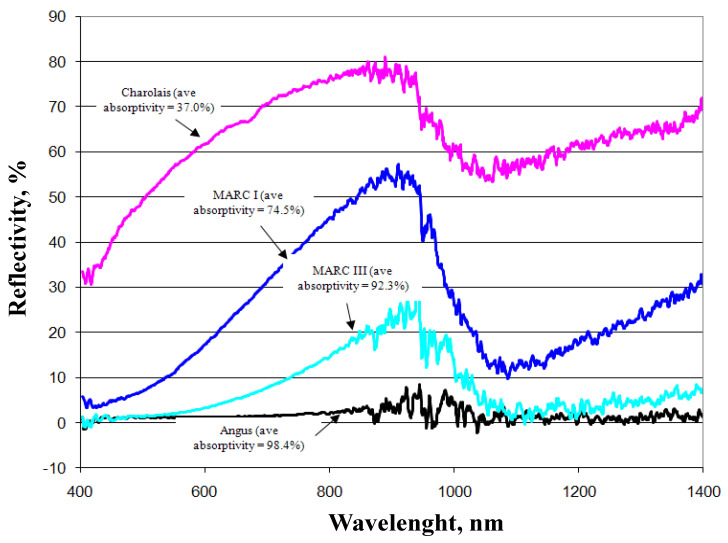
Measured percent reflectivity of the haircoat of four breeds of cows [11].

**Figure 5 animals-13-03087-f005:**
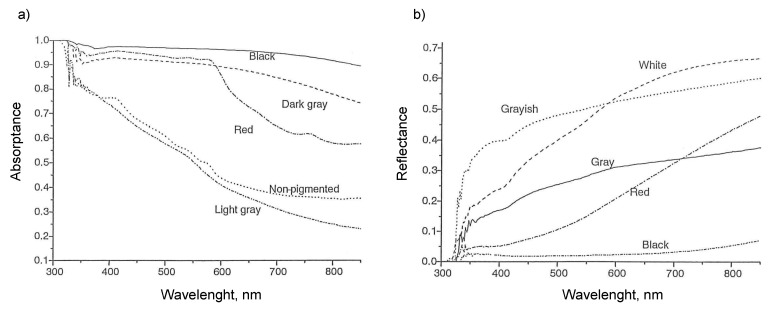
Optical properties of heifers of different hair color (**a**) absorptivity and (**b**) reflectivity as a function of wavelength [14].

**Figure 6 animals-13-03087-f006:**
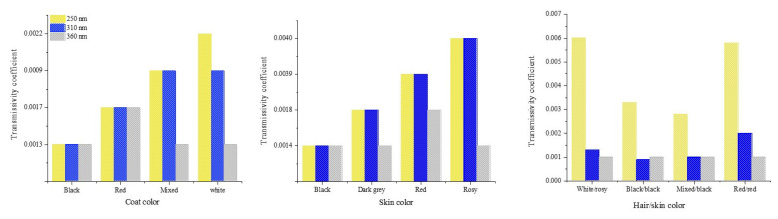
Transmissivity coefficient of different haircoats and skin pigmentations of cattle in tropical regions [1,15].

**Figure 7 animals-13-03087-f007:**
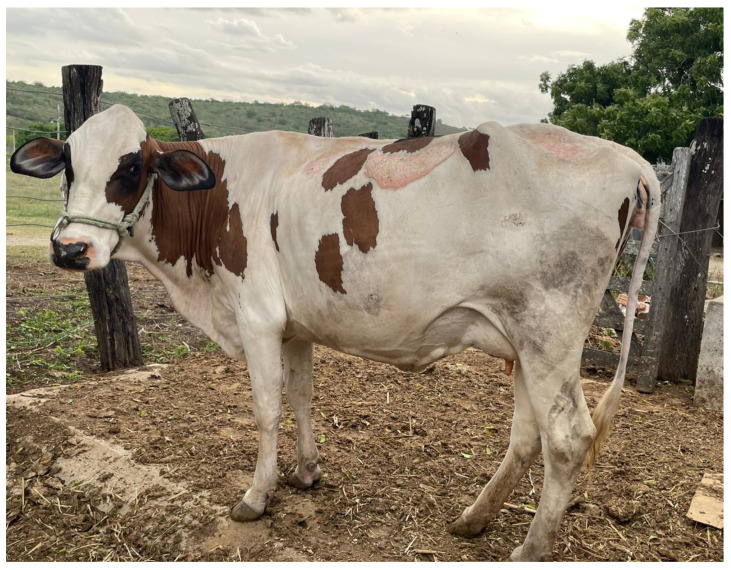
Skin burns in white spots of predominantly white Holstein cow in an equatorial tropical region of Brazil. This image is a courtesy from Dr. Vinicius F. C. Fonsêca [23].

**Figure 8 animals-13-03087-f008:**
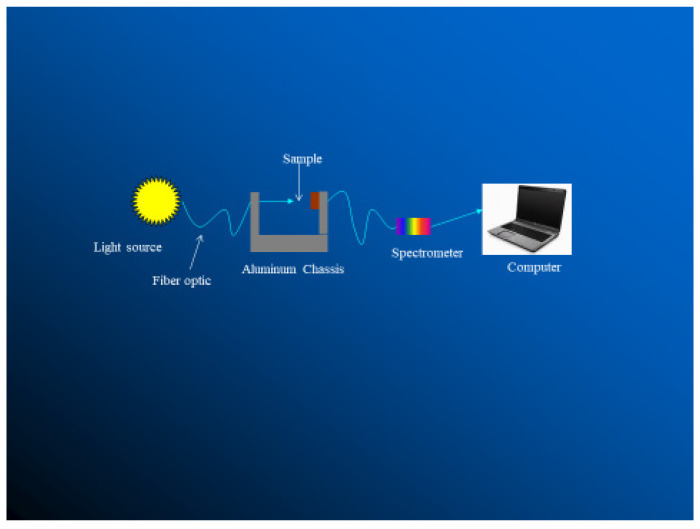
A schematic for measuring transmittance of haircoat [12].

**Figure 9 animals-13-03087-f009:**
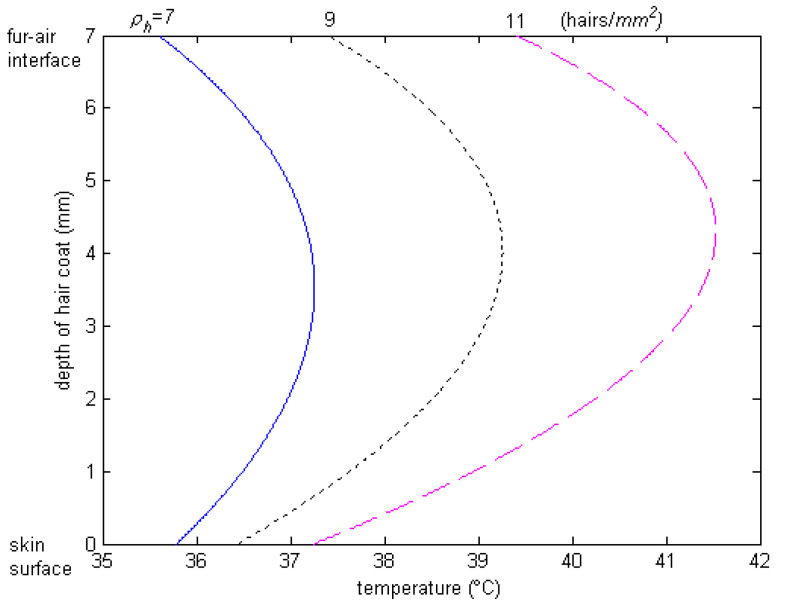
Temperature profile through black haircoat for different haircoat densities of Holstein calves. Note that peak temperature occurs deep down into the haircoat [25].

**Figure 10 animals-13-03087-f010:**
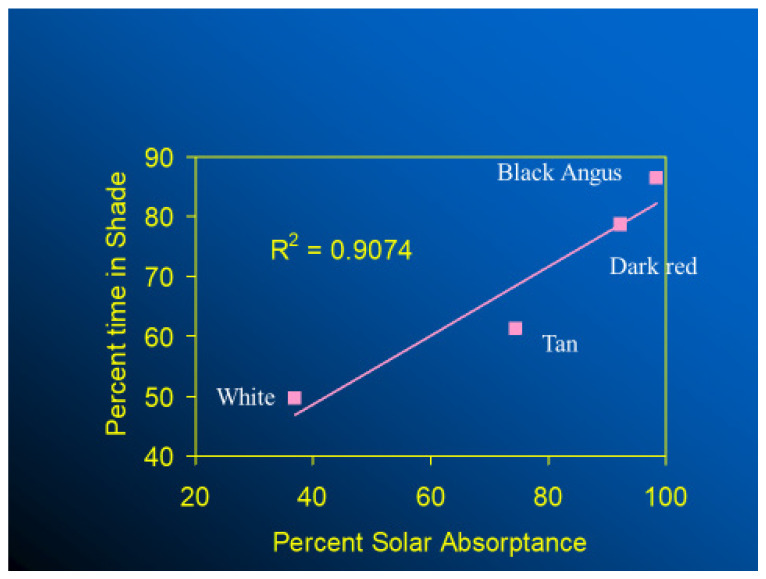
Effect of haircoat color on shade seeking [30].

**Table 1 animals-13-03087-t001:** Physical properties of hair and haircoat of cattle. Values given are mean and standard deviation (numbers in parenthesis are sample sizes).

Breed	Hair Length (mm)	Hair Diameter (μm)	Haircoat Density (#/cm^2^)	Haircoat Depth (mm)	Study Location	Comments
^1^ Holstein (Rump)	21.62 ± 2.93 (85)	48.80 ± 9.09 (21)	5200 (at skin level)	12.35 ± 1.29 (13)	USA (Wisconsin)	Winter
Holstein (Lumbar)	22.05 ± 3.69 (86)	54.97 ± 9.27 (19)	7200 (at skin level)	14.16 ± 1.08 (15)	USA (Wisconsin)	Winter
Holstein (Belly ventral)	21.04 ± 5.02 (67)	41.66 ± 10.75 (28)	6300 (at skin level)	15.28 ± 1.97 (15)	USA (Wisconsin)	Winter
Holstein (Thoracic lateral)	17.13 ± 2.68 (62)	41.73 ± 8.03 (26)	6250 (at skin level)	10.55 ± 1.41 (15)	USA (Wisconsin)	Winter
Holstein (Head)	18.40 ± 3.12 (79)	43.70 ± 7.30 (28)	3900 (at skin level)	12.56 ± 1.76 (15)	USA (Wisconsin)	Winter
Holstein (Grand total)	20.05 ± 3.49 (399)	46.375 ± 10.7 (245)	3900–7200 (at skin level)	12.98 ± 2.23 (73)	USA (Wisconsin)	Winter
^2^ Holstein	12.60 ± 3.45	62.49 ± 5.6	987.00 ± 347	2.48 ± 0.48	Brazil	Tropical climate
^3^ Braford	10.41 ± 3.91	30.98 ± 8.13	993.18 ± 503	3.73 ± 1.72	Brazil	Tropical climate
^4^ Gyr (Zebu)	4.68 ± 1.22 (15)		1140.62 ± 289 (15)	2.68 ± 0.34 (15)	Brazil	Tropical climate
Zebu x 50–75%	6.81 ± 1.81 (28)		971.62 ± 292 (28)	2.68 ± 0.47 (28)	Brazil	Tropical climate
Holstein						
Zebu x > 75%	8.74 ± 2.22 (31)		1071.90 ± 237 (31)	2.95 ± 0.44 (31)	Brazil	Tropical climate
Holstein						
^5^ Holstein	11.1	40.1	1201	2.2	Brazil	Tropical climate
Brangus	20.8	41.2	867	2.4	Brazil	Tropical climate
Nelore	13.6	53.5	1944	3.2	Brazil	Tropical climate
Simmental	15.0	39.0	939	1.7	Brazil	Tropical climate
Canchim	17.1	47.0	1206	1.9	Brazil	Tropical climate

^1^ [12]; ^2^ [3]; ^3^ [5]; ^4^ [13]; ^5^ [14]. Values are averaged for the same breed but different colors. **# refers to number**.

**Table 2 animals-13-03087-t002:** Spectral properties of haircoat of cattle.

Breed	Hair Color	Absorptivity	Reflectivity	Transmissivity	Study Location
^1^ Angus	Black	0.98			USA
MARC III	Dark Red	0.92			USA
MARC I	Tan	0.75			USA
Charolaise	White	0.37			USA
^2^ Holstein	Black	0.89 ± 0.01			USA
Holstein	White	0.657 ± 0.045			USA
^3^ Holstein	Black *	0.902	0.0893		Brazil
Holstein	White *	0.518	0.4536		Brazil
^4^ Holstein	Black	0.93	0.06	0.01	Brazil
Holstein	White	0.43	0.53	0.04	Brazil
Holstein	Red	0.37	0.44	0.19	Brazil
Brangus	Black	0.92	0.07	0.01	Brazil
Nelore	Dark Grey	0.91	0.04	0.05	Brazil
Simmental	Red	0.54	0.29	0.17	Brazil
Canchim	Gray	0.27	0.66	0.07	Brazil

^1^ [11]; ^2^ [21]; ^3^ [22]; ^4^ [14]. * values here refer to effective absorptivity and reflectivity.

**Table 3 animals-13-03087-t003:** Least-square means of physical properties of haircoat of Holstein cows for different month of sampling, cow age, and coat color (n = number of samples, l = haircoat depth, L = hair length, *ρ*_h_ = haircoat density, D = hair diameter) [22].

Effects	n	l (mm)	L (mm)	*ρ*_h_ (Hair/cm^2^)	D (µm)
By month of sampling					
November	771	2.56 ± 0.024	12.08 + 0.16	1062 ± 21	61.56 ± 0.27
December	454	2.50 ± 0.026	12.90 ± 0.18	1041 ± 23	62.41 ± 0.31
January	122	2.78 ± 0.046	13.82 ± 0.32	975 ± 41	61.07 ± 0.55
February	158	2.63 ± 0.036	12.35 ± 0.25	1222 ± 32	60.50 ± 0.43
March	332	2.56 ± 0.032	15.16 ± 0.22	1281 ± 28	59.53 ± 0.38
April	109	2.55 ± 0.051	17.98 ± 0.35	1070 ± 45	59.53 ± 0.60
By cow age (years)					
<2	152	2.53 ± 0.074	14.27 ± 0.52	1316 ± 65	57.61 ± 0.88
2–3.5	684	2.55 ± 0.049	13.39 ± 0.34	1163 ± 43	59.75 ± 0.58
2.5–5	530	2.66 ± 0.043	14.14 ± 0.30	1060 ± 37	60.38 ± 0.51
5–6.5	301	2.78 ± 0.040	15.29 ± 0.28	1073 ± 35	60.39 ± 0.47
6.5–8	184	2.55 ± 0.052	13.78 ± 0.36	987 ± 46	62.20 ± 0.62
>8	95	2.50 ± 0.075	13.42 ± 0.52	1056 ± 66	64.26 ± 0.89
By coat color					
Black	973	2.40 ± 0.024	12.97 ± 0.16	921 ± 21	62.40 ± 0.28
White	973	2.79 ± 0.024	15.13 ± 0.16	1296 ± 21	59.13 ± 0.28

## Data Availability

The data presented in this study are available on request from the corresponding author.

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
