# Peer review of "Methods, Thermodynamic Applications, and Habitat Implications of Physical and Spectral Properties of Hair and Haircoats in Cattle"

_animals, 2023, doi:10.3390/ani13193087_

Round 1

Reviewer 1 Report

I believe readers would be interested in the content of this review article and that it would be well received after a significant restructure of the body of text. The content and intent is good, however as it is, it is hard to read as a complete body of work, and does not read like a review.

Highlights

The text in this section does not represent the ‘highlights’ of this review. The focus of this review is on methodology, and it would perhaps be better if the authors highlight the best method(s) of hair coat measurements in this section, leading with the last sentence of this section.

Line 11 – By “Black Holsteins” do the authors mean back-and-white Holsteins? This should be clarified to avoid confusion.

Simple Summary

This section reads more like an introduction rather than a summary of the review. Also, I understand the point the authors are trying to make by talking about polar bears and goats, but the review is about hair and haircoat properties in cattle so this should be revised and replaced with examples specific to cattle.

Line 20: What is meant by "a thermal point"? Best to clarify this.

Abstract

Line 33/34 states that this is a "study to establish accurate and consistent measurements of these properties", however this is a review article - not a study. Furthermore, the authors are not "establish[ing] accurate and consistent measurements", but rather reviewing what has been reported in the literature.

At the end of the abstract the authors write about thermoregulation, shade seeking behaviour, and the benefit of having a particular haircoat colour in different ecological habitats. The literature on these topics is not well discussed anywhere in the larger body of text. I think the review would greatly benefit from reviewing literature on how coat traits influence thermoregulation, including their involvement in heat stress. This would also provide a better introduction to why the study of physical and spectral properties of the coat are important.

Introduction

Line 51: The authors have provided an example of what "low coat thickness" is but should also provide an example of a thick coat for comparison e.g., hair coat thickness in Highland cattle.

Line 65: It is important here to make a distinction between cattle native to a region that are well suited to their environment, versus cattle that have been imported/selected on for production value. For example, Holstein cattle are also found in Africa and farmed for milk production, but they are not adapted or suited to the climates in those areas. Also, coat colours can be influenced by cultural preferences, with little consideration for how well the coat colour is suited for the environment.

Line 74: The authors write about “present-day high milk volume-producing cows” but do not discuss in any detail the relevance of coat characteristics to this. Perhaps they could consider reviewing literature that links coat characteristics to thermoregulation and changes in production, fertility, and immune function.

Line 76: Again, this a review, not a study

Line 78: What constitutes as "recent" literature?

Perhaps the authors would consider using their objectives as a guide to write their ‘Simple Summary’ and ‘Highlights’ sections.

The authors could also consider talking about heat stress, the physiological impacts of heat stress, and how heat stress has historically been studied (i.e., talking about the temperature humidity index; THI). This would provide good background and an easy segway into why they are interested in studying hair and hair coat in cattle. It would also provide sufficient background for the authors to talk about "heat exchange models, and their implications on voluntary thermoregulation of cattle by seeking shade" as eluded to in the abstract, but not clearly discussed anywhere in the main body of the text.

Literature Review

This section would benefit from more in-depth discussion. It would be beneficial if the authors could discuss the methods used to measure hair coat properties for each of the papers presented and discuss advantages and disadvantages of these methods.

Line 208: Reference is in a different font

Line 235: ‘Lighter’ hair coat rather than ‘brighter’ hair coat? 

Measurement procedures

There is a lack of referencing here and this section does not fit the rest of the paper. It would perhaps be best to merge this into the literature review with the studies they belong to, and have one paragraph at the end and discusses the ‘best approach’ and why it is the best approach, or something along the lines of "based on the literature we propose that future studies aiming to measure hair coat properties should ..."

Habitat implications of hair and haircoat physical and spectral properties

I understand the intention behind discussing coat characteristics in Polar bears and goats, however this paper is about cattle and would benefit from examples relating directly to cattle. I would suggest removing all references to polar bears and goats and referencing cattle-based studies that demonstrate coat adaptations to hotter climates. The authors could also use this section to discuss shorter hair coats and tick resistance in cattle. The following publications could be used as a starting place for these discussions:

Katiyatiya, C. L. F., G. Bradley, and V. Muchenje. "Thermotolerance, health profile and cellular expression of HSP90AB1 in Nguni and Boran cows raised on natural pastures under tropical conditions." Journal of Thermal Biology 69 (2017): 85-94.

Littlejohn, Mathew D., et al. "Functionally reciprocal mutations of the prolactin signalling pathway define hairy and slick cattle." Nature communications 5.1 (2014): 5861.

Marufu, Munyaradzi C., et al. "Relationships between tick counts and coat characteristics in Nguni and Bonsmara cattle reared on semiarid rangelands in South Africa." Ticks and tick-borne diseases 2.3 (2011): 172-177.

The body of the text needs review as there are small grammatical errors, words missing, and incorrect tense. The "Highlights" and "Simple Summary" require more extensive editing as the errors throughout these portions of text make the text difficult to read.

Author Response

Animals-2498 962
Responses to Comments from Reviewer 1:

We sincerely appreciate the time Reviewer 1 spent reviewing the paper and for his appropriate and
detailed comments. As a result of his comments, the paper is much stronger and more organized. The
following are our responses to the Reviewers comments:

(1) We have addressed all the editorial changes suggested.

(2) The paper is restructured. It is now made to look like a review paper.

(3) The Simple Summary and Conclusions sections have changed and now refer to the properties of
hair and haircoat.

(4) The measurement methods and procedures are now merged into the literature review as
suggested by the reviewer.

(5) Five additional references including the three suggested by the reviewer are added.

(6) The discussion about Polar bears and black goats is completely removed from the Simple
Summary section but is left in the Habitat Implication section to highlight, as an example, the
effect of color on solar absorptivity and reflectivity.

(7) Methods of measurements for physical and spectral properties of hair and haircoat are given in
the cited papers.

(8) present-day high milk volume-producing cows” is referenced.

(9) We did not find references that relate the benefit of having a particular haircoat color with respect
to shade seeking behavior other than the study (ours) cited in this paper.

Comments not addressed:

(1) Comment: Perhaps they could consider reviewing literature that links coat characteristics to
thermoregulation and changes in production, fertility, and immune function.

Response: It is not the scope of this paper to address “ changes in production, fertility, and
immune function” due to coat characteristics and frankly we don’t know of any study to those
issues.

(2) Comment: The authors could also consider talking about heat stress, the physiological
impacts of heat stress, and how heat stress has historically been studied (i.e., talking about
the temperature humidity index; THI).

Response: The focus of this paper is on measurement methods and procedures of physical
and spectral properties of hair and haircoat not on heat stress of cattle which is a big topic in
itself.

Reviewer 2 Report

This manuscript is really a literature review. It should be submitted via the "Animals" Literature review portal.  The information is valuable but really represents research to date on a poorly studied topic.  The information in the manuscript is important and the techniques used to assess the thermal properties of "hair coat" is needed by the scientific community.  I do think you have discussed the anomalies of "white polar bears" and black goats/Holsteins too much in respect to the message of the literature review.  You do need to mention anomalies but not make the a focus of your "simple summary and conclusions.  Also, please find additional papers on this topic.  The literature review is missing recent papers and could rely more on thermodynamic models used to estimate nutrient requirements or at least discuss the interaction of thermoregulation and nutrient requirements and intake.

Author Response

Response to Comments from Reviewer 2:
We thank and appreciate Reviewer 2’s positive comments. The Reviewer said: “The information in the manuscript is important and the techniques used to assess the thermal properties of "hair coat" is needed by the scientific community.

(1) The paper is restructured. It is now made to look like a review paper. The word “this study” is changed to “this literature review”.

(2) We removed the discussion on Polar bears and black goats from the Simple Summary and Conclusion sections as suggested.

(3) We have added five additional references.

(4) It is not the focus of this paper, however, to discuss “thermodynamic models used to estimate nutrient requirements or the interaction of thermoregulation and nutrient requirements and intake.”

Reviewer 3 Report

The paper was gathering information on Hair and Haircoat in Cattle, measurement methods, thermodynamic applications and habitat implications of Physical and Spectral Properties. This data is deemed important because it pulls information together that can be used to guide or inform farmers on the type of breeds they can chose depending on the continent and the dominant climatic condition. It also guides on methods that can be used by different authors to continue to research on hair and haircoat in cattle.  

The highlight was a bit weak because it was supposed to highlight key finding or conclusion. See attached document for more comments

The language also need moderate editing because a lot of short sentences appeared without any flow. In other sections, the methods didn't point out the same principles and no indication on differences. It was very hard to really look at what authors where trying to confirm in terms of the method or even agree on which method was best according to their review. 

The conclusion was also describing instead of stating the key finding from the different papers reviewed or straight to the clear observation.  The reading was expecting to see which method was best to study hair and haircoat in cattle and why? which animals will be best in which climate and the reason. The rest of my comments are in the attachment.

The language also need moderate editing because a lot of short sentences appeared without any flow. There were some places where present these was used instead of past tense during reviewing or gathering of the data. All of that is indicated in the paper attached. However, if a bit of effort is put in, the paper should be good to go.

Author Response

Response to Comments from Reviewer 3:

We thank and appreciate Reviewer 3’s positive comments. The Reviewer said: “ This data is deemed important because it pulls information together that can be used to guide or inform farmers on the type of breeds they can chose depending on the continent and the dominant climatic condition. It also guides on methods that can be used by different authors to continue to research on hair and haircoat in cattle.
(1) The Simple Summary and Conclusions sections are changed and now refer to the properties of hair and haircoat.

(2) We have addressed the editorial changes of the short sentences.

(3) We have added five additional references.